# Attachment Stability and Longitudinal Prediction of Psychotic-like Symptoms in Community Adolescents over Four Months of COVID-19 Pandemic

**DOI:** 10.3390/ijerph20166562

**Published:** 2023-08-11

**Authors:** Cecilia Serena Pace, Stefania Muzi, Wanda Morganti, Howard Steele

**Affiliations:** 1Department of Educational Sciences, University of Genoa, 16128 Genoa, Italy; stefania.muzi@edu.unige.it (S.M.); wanda.morganti@edu.unige.it (W.M.); 2Center for Attachment Research, New School for Social Research, New York, NY 10011, USA; steeleh@gmail.com

**Keywords:** Friends and Family Interview, adolescence, attachment, test–retest, psychometric study, psychotic symptoms, psychotic-like experiences, dissociation, hallucinations, Youth Self-Report

## Abstract

*Background:* The Friends and Family Interview (FFI) is assumed to be a valid method to study attachment stability and attachment-related psychopathological processes in adolescence, but no studies have yet tested the test–retest reliability of this interview or the longitudinal association of attachment patterns in response to the FFI from adolescents with symptoms such as psychotic-like experiences (e.g., hallucinations, bizarre behavior, dissociation, self-harm) that are known to have increased during the COVID-19 pandemic. *Methods:* This study involved 102 community adolescents (*M* = 14.64, *SD* = 1.63, 46% males) assessed twice: during a severe COVID-19-related lockdown (in Italy) (T1) and four months later (T2). Measures were the FFI (assessing attachment patterns: secure-autonomous, insecure-dismissing, insecure-preoccupied, and insecure-disorganized) and the thought problems scale of the Youth Self-Report to assess psychotic-like symptoms. *Results:* revealed high stability of four-way attachment classifications over four months (93.5%), with a modest yet significant link between higher disorganization at T1 and higher scores of thought problems at T2, *p* = 0.010. *Conclusions*: The FFI shows high test–retest reliability and can be a valid, age-adapted option to assess adolescents’ attachment. Attachment disorganization should be further investigated as possibly related to psychotic-like experiences in community adolescents.

## 1. Introduction

The strength of attachment theory [1] lies in the way it led to the development of assessment procedures capable of capturing attachment patterns in different age groups. For instance, both the Strange Situation Procedure (SSP) for infants [2] and the Adult Attachment Interview (AAI) [3] for adults are considered gold standard measures supporting empirical studies and the translation of attachment theory into clinical practice [4,5,6]. 

However, the study of attachment and its correlates is still a challenge for scholars interested in adolescence, a phase of life characterized by sudden and marked changes that can also create discontinuity in the attachment system [7,8,9]. Indeed, it is also difficult to establish a gold standard measure of attachment in this age of development where attachments are changing, often in flux and sometimes volatile, calling for a psychometric investigation of the test–retest reliability and concurrent validity of attachment measures in adolescence with psychopathological symptoms, so fundamental to attachment theory [9].

### 1.1. Questioned Stability of Adolescents’ Attachment and Assessment Methods

As postulated by Bowlby [1], during the first year of life, a child establishes an attachment bond(s) with the preferred primary caregiver(s), i.e., attachment figure(s). From repetitive experiences with attachment figures, the infant child develops mental schemes of how to behave and express the needs for proximity and autonomy in response to a felt understanding of what to expect from attachment figures familiar to the child. Children’s schemas for self-other relationships will depend on how the ‘other’ behaves toward the child, especially when the child is distressed. Bowlby [1] called these mental schemes Internal Working Models (IWMs) [1] of attachment (i.e., attachment representations), postulating that children generalize the conclusions felt from important repeated interactions into abstract attachment representations that include rules for how to behave with different attachment figures, and these largely unconscious ‘rules’ guide behavior far beyond primary relationships into relationships with peers, teachers, and others across the lifespan. Therefore, the extreme attachment theory hypotheses regarding early attachment representations are that they may influence how people process and perceive emotionally related information and organize their behavior within all significant relationships from infancy onward. Where such continuity is observed, it may be not only because of early attachment experiences but also because of how early attachment experiences tend to repeat themselves and become consolidated as characteristics of the person. Specifically, infants can and do demonstrate different attachment patterns depending on how they are treated by attachment figures. When IWM includes the expectation that the attachment figure is available and responsive to the child’s attachment needs, the child–parent attachment pattern can be classified as secure. When the IWM is deemed to contain an expectation of the attachment figure as unavailable or unpredictable, the person shows one of the insecure patterns [10] that can be classified as avoidant (dismissing)—minimizing the expression of attachment needs of closeness and comfort-seeking to avoid fear of parental rejection; ambivalent (preoccupied)—exaggerating attachment behaviors to require comfort from a parent probably perceived as inconsistent in responding to the child’s attachment needs; and disorganized (unresolved)—a consequence of adverse and potentially traumatic experiences with primary caregivers and/or excessive turnover of caregivers in infancy, for instance, in cases of early institutionalization [2,3].

Bowlby theorized [1] that infant attachment representations evolve from pre-verbal to representational levels as the child’s capacities mature, the social context widens, and cognitive capacity evolves [11]. These representations tend toward stability in a stable environmental context, but they can also be restructured to facilitate adaptations during marked life transitions [1]. An example of this process occurs during adolescence, a phase characterized by significant physical-pubertal, cognitive, relational, and behavioral changes, including modifications in attachment relationships and behaviors, i.e., seeking autonomy from parents and starting to consider peers as potential attachment figures, that prompt a restructuring of attachment representations [12,13]. Indeed, meta-analytic evidence [8] suggests adolescence as a possible transition point where attachment shows relative discontinuity with infant attachment but more continuity with adult attachment than with prior infant–parent attachments. Thus, studying adolescents’ attachments provides an avenue to further understand processes of stability and change in attachment patterns.

Moreover, the issue of how to assess attachment during adolescence is still debated [9]. Indeed, with adolescents, the most widely used assessment methods are self-report questionnaires [14] that, although rapid and effective, appear to be less efficient than interviews or projective methods in capturing insecure, especially insecure-disorganized, patterns [15,16]. A systematic review [9] recommends the Child Attachment Interview (CAI) for children aged 7–13 years and the Adult Attachment Interview (AAI) for adolescents aged over 15, but both focus only on representations of parental relationships, missing the transition to new (peer/romantic) attachment relationships during adolescence.

An alternative option is the Friends and Family Interview (FFI) [17,18], an age-adapted semi-structured interview for adolescents. Like the AAI, an outcome of the FFI is the best-fitting overall attachment category. Specifically, the secure-autonomous category is assigned when the narrative is mostly coherent according to Grice’s conversation maxims (quality, quantity, relation, manner) [19] and the person shows a value for attachment. An insecure category is assigned when the narrative coherence lacks for specific reasons to do with defensive affective regulation strategies emerged according to the prevalent pattern, e.g., low coherence in terms of quality and/or relation due to discrepancies between positive parental descriptions and an obvious absence of episodic memories (i.e., idealization), in often too-brief insecure-dismissing interviews, or excessive quantity or rushed speech that seems overly angry and inappropriate as seen in interviews judged insecure-preoccupied, or, finally, responses that are lacking in all aspects of narrative coherence due to the intrusion of potentially unresolved traumatic memories in the case of interviews judged disorganized. Compared to the AAI or the CAI, the FFI has a number of advantages insofar as it permits researchers to consider all four principal attachment patterns, including the most prominent one observed, plus a wide range of relevant adolescent life domains (including relationships with peers, favorite teachers, parents, and siblings). In other words, the FFI employs a dimensional coding system where each pattern is rated on a scale, and the highest score corresponds to the best-fitting attachment category, providing both a categorical and a dimensional outcome concerning attachment patterns. Therefore, the FFI is a promising assessment method, but at the time of the past cited reviews (refs), the psychometric properties of the FFI were still scarcely documented, while recent research indicates an increased effort to deepen knowledge of the psychometric properties of the FFI [17,18,19,20,21,22,23,24]. Specifically, the validity of the FFI has been demonstrated in different studies [17,20,21,22,23,24,25,26]. Convergent validity has been supported by studies that found that FFI scores/classifications responses in children 11 and older were predicted by pre-birth [17] and contemporary [20] parents’ Adult Attachment Interview (AAI) responses. As well, FFI outcomes were predicted by SSP data of the same adolescent participants when they were infants [17] and also by a study that showed that the FFI distribution overlapped with the AAI meta-analytical distributions in community adolescent populations [21]. Further, the coherence in the FFI showed discriminant validity based on middle-aged children’s verbal intelligence [17], and patterns and coherence resulted independently of emotional–behavioral problems and cognitive–verbal abilities of adolescents [21], as well as EEG responses [22]. Moreover, the FFI shows concurrent validity with attachment questionnaires [23,24], content validity with a factorial structure [25], and adequate inter-country consistency in Belgium and Romania [26].

One FFI psychometric property not yet investigated is test–retest reliability, which is vital to consider given the widely reported significant changes adolescents undergo regarding their need to redefine relations with parents and establish new social ties with friends, siblings, and teachers, all of which are inquired about by the FFI. Briefly, test–retest stability may or may not be as strong as attachment measures indicate for other less turbulent developmental periods. Studies examining the test–retest reliability of other attachment measures revealed 69% concordance of attachment classifications assigned to children aged 7–12 with the Child Attachment Interview over three months [27], 71% concordance over 4 years employing the Attachment Interview for Childhood and Adolescence (AICA) in participants aged 10–14 years [28], and 78% for the Adult Attachment Interviews of adults aged 19–33 after 2 months [29].

### 1.2. A Focus on Association between Attachment and Psychotic-like Experiences among Adolescents

Another longstanding line of research in the attachment literature investigates links between attachment and youths’ adjustment and mental health [16,29]. On one hand, attachment security is largely supported as a life-long protective factor against psychopathology [30,31], while allied meta-analytic evidence [15] supports connections between adolescents’ insecure-dismissing and insecure-preoccupied patterns and internalizing problems (i.e., anxious and depressive symptoms) and between disorganized attachment and both internalizing and externalizing symptoms, i.e., aggressive and oppositive-defiant behaviors.

Beyond a traditional focus on internalizing and externalizing problems, attachment scholars may be interested in deepening connections with other adolescents’ psychopathological symptoms, such as *psychotic-like experiences* (PLEs) [32], namely “psychotic symptoms in the absence of illness” [33], referring primarily to visual and auditory hallucinations, bizarre behavior, and delusions [33,34,35,36].

From a clinical perspective, PLEs interest mental health professionals and researchers because they are experienced by 40 to 66% of community adolescents [36,37], and these symptoms during adolescence indicate a general psychopathological vulnerability, being predictive of severe mental health disorders later in life [38], including adult psychosis and dissociation [38,39]. Moreover, PLEs are often comorbid with other internalizing and externalizing problems, interfering with the prognosis and treatment of other mental health disorders [38,40]. Therefore, research on correlates of PLEs in adolescents is recommended [33], especially after the prolonged self-isolation during the COVID-19 pandemic, where PLEs in adolescents dramatically increased [40].

In attachment research, PLEs—considered dissociative experiences, i.e., part of a dissociation process [41]—were investigated in conjunction with measures of attachment disorganization [41,42,43,44,45,46]. Specifically, attachment studies connect infant attachment disorganization to adolescent and adult dissociation [44,45,46,47] and psychotic symptoms or disorders [41]. A meta-analysis [41] also reports associations between psychotic symptoms and both dismissing and preoccupied patterns in both clinical and non-clinical samples but includes few studies on adolescents. The scarce attachment literature on the topic suggests that higher attachment preoccupation (in terms of anxiety) predicted more PLEs in non-clinical community adolescents [32]. Other studies report associations between attachment dismissal, preoccupation, and PLEs among both non-clinical and clinical adolescents [48,49], as assessed with the thought problems scales of the Youth Self-Report (YSR) [50]. This work is limited insofar as all these studies are cross-sectional, not providing information on the longitudinal effect of attachment on adolescents’ PLEs, and most of these past studies used attachment questionnaires [32,48,49], none of which assessed attachment disorganization. Therefore, from an attachment perspective, investigating the associations between adolescents’ attachment patterns and PLEs in non-clinical adolescents could further knowledge regarding the continuity of negative sequelae of attachment disorganization over the lifespan, e.g., is it the case that only infant disorganization predicts later disorganization [47]? Or might disorganization in response to the FFI be uniquely linked to increased vulnerability to dissociation, or if, in a period of general attachment restructuration such as adolescence, organized insecure patterns such as dismissing and preoccupation can increase vulnerability to dissociative experiences.

### 1.3. The Current Study

From the above summary of published work, including the Friends and Family Interview, or FFI, some notable gaps are evident in the FFI literature. First, the recent effort of researchers to contribute to the knowledge about the psychometric properties of the FFI [17,18,19,20,21,22,23,24,25,26] has not yet addressed the topic of the test–retest validity of this interview, whose investigation would also increase the knowledge about processes of continuity or discontinuity in attachment during the critical adolescent period of development. Therefore, the first goal of this study is to investigate the stability of adolescents’ attachment representations assessed by measuring the test–retest reliability of the FFI categories over four months, hypothesizing stability of attachment classification in this brief period as in other studies employing other interviews with adolescents [27,28,29].

Second, the dramatic increase in PLEs of adolescents after the COVID pandemic calls for further investigation efforts on possible correlates of these potentially psychopathological symptoms, and so far the attachment literature has suggested that attachment insecurity may be related to dissociative experiences in adults [42,43,44,45] and PLEs in adolescents [30,45,46], but longitudinal and interview-based studies are still few in number, and no studies investigated associates of disorganization in response to the FFI. Therefore, a second goal of this study is to examine the longitudinal associations between attachment patterns, namely secure, insecure-dismissing, insecure-preoccupied, and insecure-disorganized, when compared with PLEs reported on by non-clinical community adolescents, and the FFI will permit consideration of different types of insecure patterns, providing a more comprehensive picture of adolescent adaptation than is available from questionnaires [15]. Based on existing findings on adolescents, higher levels of PLEs were expected to be reported from those adolescents with higher scores for insecure (preoccupied or dismissing) patterns, and especially from those adolescents with higher disorganization scores in response to the FFI. Such expectations are consistent with past findings regarding PLEs concerning adults [32,41,42,43,44,45,46,47,48,49].

## 2. Materials and Methods

### 2.1. Participants and Procedure

This brief-longitudinal research procedure received approval by the Ethics Committee of the Department of Educational Sciences of the University of Genoa, protocol n. 037, and complied with the ethical guidelines of the worldwide scientific communities.

The data collection took place during the first wave of the COVID-19 pandemic. The first data collection was in April–June 2020 (T1), when in Italy there was a severe lockdown, so all interpersonal contacts were restricted, and all activities were translated into an online form. The second data collection was four months after T1, August–October 2020 (T2), when Italy underwent intermittent lockdowns with frequent passages from online to in-person school and activities. For this reason, all research activities have been performed in digital form since recruitment. 

Potential participants for the entire longitudinal research were recruited through public high schools in Ligurian region after formal agreement with the directors. Inclusion criteria were age between 12 and 17 years old and the absence of diagnoses for psychiatric disorders and physical or intellectual disabilities. The research team illustrated the research and content of informed consent to adolescents and their parents through letters. Parents signed an informed consent form for their adolescents’ voluntary participation. 

Of the one hundred and twenty-five adolescents contacted, twenty-two refused (18% attrition), and one hundred and three adolescents agreed to participate in the research and completed measures of T1. Of them, participants can be included in this study if they responded to the attachment interview FFI and the measure for psychotic-like symptoms in both data collections. Therefore, this study T1 includes 102 community adolescents aged 12–17 years old (45% males; Mean age (M_age_) = 14.64, Standard Deviation (SD_age_) = 1.63), coming from low-risk families of medium (77%) or high (12%) social educational level. 

Nine adolescents dropped out before T2 (six were impossible to retrieve although contacted three times, and three refused to participate at the second step without explaining, as guaranteed by the informed consent), so T2 included ninety-three adolescents aged 12–17 years (43% males; M_age_ = 14.90, SD_age_ = 1.65). Inspection of demographic data was undertaken to see if there were differences between the sample in T1 and the subsample who completed T2, and there were no significant differences in gender, family SES distribution, or age (all *p* > 0.152).

Six trained MSc students in Clinical Psychology contacted participants and administered the FFI interview online in a video call, paying attention to the privacy guarantee and audio-recording the interview for further verbatim transcription necessary for coding. After the interview, the adolescent filled out individually an online survey received at the school email address with the online assistance of the MSc student. Each survey was previously matched with the participant through a token, which was also used to save the audio record of the FFI. This procedure was used both in T1 and in T2 (four months after T1 for each participant), and each session lasted around 1 h and a half. After the transcription, two certified coders (the second and third authors) rated the FFIs. Both raters independently and blindly double-rated 17 interviews of T1 (17%) and 15 interviews of T2 (16%), who reached 100% agreements on 2-way, 4-way, and organized–disorganized categories.

### 2.2. Measures

#### 2.2.1. Attachment

The Friends and Family Interview V.5 [17,18,22] has been used to assess adolescents’ attachment. This audio- or video-taped 28-question interview assesses several domains of attachment (i.e., coherence, reflective functioning, self-esteem, relationship with the best friend and with siblings, affective regulation strategies, and differentiation of parental representations) with subscales for each domain. After *ad verbatim* transcription, certified raters assign 1–4 scores in each subscale, where 1 = no evidence, 2 = a little evidence, 3 = moderate evidence, and 4 = marked evidence. According to scoring guidelines, combinations of scores in the domains guide the assignment of a score 1–4 in each of the four patterns: secure-autonomous (S/F), insecure-dismissing (DS), insecure-preoccupied (P/E), or insecure-disorganized (D), allowing a dimensional evaluation of each attachment pattern. The highest score on these patterns’ scales corresponds to the best-fitting attachment category at a categorical level. In this study, Cronbach’s alphas were 0.86 in T1 and 0.87 in T2.

As basic assumptions for the test–retest reliability, possible differences in FFI outcomes due to the interviewer (*n* = 6) or rater (*n* = 2) were preliminarily checked. The FFI secure–insecure category assigned did not vary according to the interviewer (T1, *X*^2^(5) = 9.80, *p* = 0.081, *Cramer’s V* = 0.081; T2; *X*^2^(5) = 8.99, *p* = 0.011, *Cramer’s V* = 0.011). FFI outcomes did not change according to the rater, as raters reached 100% inter-rater agreement in all categories (*k* =1), and pattern scores assigned to double-coded interviews by the two raters significantly correlated with each other, all *p* < 0.018.

#### 2.2.2. Psychotic-like Experiences

The thought problems scale of the widely known self-report questionnaire Youth Self-Report 11–18 years [50] has been used to assess psychotic-like symptoms in teenagers. This scale is the sum of seven items (9, 40, 65, 69, 82, 83, 84) where the adolescent reports unusual experiences such as visual or auditory hallucinations, bizarre ideas, and behaviors. This scale has been established as a good first-step screening measure to detect prodromic signs of psychotic disorders [50]. As it is composed of only 7 items, this scale has often shown Cronbach’s alpha values lower than 0.70 [51,52]. In this study, Cronbach’s alpha was 0.67, and there was good item inter-correlation (all values > 0.30, *p* < 0.000), indicating acceptable reliability for a scale composed of few items [53,54,55]. The Cronbach’s alpha for the YSR seven-item scale was 0.93.

#### 2.2.3. Demographic Information

Information about participants’ demographics, education, presence of diagnoses, and family (e.g., structure, social–economic status of parents, and presence of siblings) was collected ad hoc [22].

### 2.3. Data Preparation and Analytic Plan

The FFI category distribution and descriptives (Mean (M) and Standard Deviation (SD)) have been reported for all measures, with statistical significance for all hypotheses, with the tolerable limit of type-one errors being set at the conventional *p* < 0.05.

After checking and managing outliers according to the criteria suggested by Hoaglin et al. [56], the two main assumptions for the use of parametric tests were tested, i.e., homogeneity of variances (by Levene’s tests) and normality of the distribution (by Shapiro–Wilk tests). Given that the data were not normally distributed, non-parametric tests were employed [57], as they do not require the assumption of normality of the population from which the sample is drawn and are appropriate for the analysis of dependent variables on dichotomous, categorical, and ordinal scales, as in this pilot study. 

As preliminary analyses, inter-rater agreement on the FFI classification was checked through Cohen’s *k*, and chi-square tests were used to check the effect of the interviewer (see above, Section 2.2.1) and of participants’ gender on FFI secure–insecure category distributions in T1 and T2, as well as to compare such distributions to the pre-pandemic one [22] as indicative of the general population distribution. The U-Mann–Whitney test was used to compare the age of participants classified as secure or insecure and the Kruskal–Wallis test to check differences among four categories, employing Kendall’s tau-b correlation coefficient (*τ_b_*) to check age differences in the T2 YSR/thought problem’s score.

For the first aim, the test–retest reliability of the FFI categories was tested using *k* di Cohen to verify the association between T1 and T2 FFI secure–insecure distributions, and *τ_b_* was used to inspect relationships between patterns and coherence scores in the two data collections, with the effect considered as: ≥0.06 weak, ≥0.26 moderate, ≥0.49 strong, and ≥0.71 very strong [58].

For the second aim, the effect of the FFI category on YSR/thought problems’ scores in T2 was checked through the U-Mann–Whitney test (two-way and organized–disorganized systems) and the Kruskal–Wallis test for the four-way system, also checking Kendall’s tau-b correlations between FFI patterns and coherence scores in T1 and YSR/thought Problems in T2.

## 3. Results

### 3.1. Descriptive Data and Preliminary Analyses

In T1, the distribution of the FFI categories of 102 adolescents was as follows: 75.5% were secure (*n* = 77) and 24.5% were insecure, of whom 22.5% were Ds (*n* = 23), and the remaining 1% were P/E (*n* = 1) and 1% were D (*n* = 1). In T2, four months later, the distribution of the FFI categories among 93 adolescents was as follows: 69.9% were secure (*n* = 65) and 30.1% were insecure, of whom 28% were Ds (*n* = 26) and the remaining 2.1% were P/E (*n* = 2). None were classified as D.

Means and Standard Deviations for all measures are reported in Table 1 (below).

Given the paucity/absence of FFI transcripts assigned to P and/or D categories in both T1 and T2, analyses on three- and four-way distributions could not be performed (i.e., *X^2^* comparisons and Kruskal–Wallis test), and the following analyses on categorical data were performed considering the FFI secure–insecure (including Ds, P, and D) distribution.

*Gender and age differences.* No gender differences were revealed in the FFI secure–insecure category distribution in T1 or T2 through the chi-square test, *p* > 0.071, but males showed lower scores for security than females both in T1, *U* = 1643.5, *p* = 0.014, and in T2, *U* = 1296, *p* = 0.040, also showing higher scores in the dismissing pattern at T1, *U* = 965.5, *p* = 0.022. 

Participants classified as secure were older than insecure ones in T1, *U* = 634.5, *p* = 0.033, and T2, *U* = 587.5, *p* = 0.013. 

The T2 YSR/thought Problem’s score was not correlated to gender (*p* = 0.38) or age (*p* = 0.31).

*Correspondence with pre-pandemic FFI data*. The FFI category distribution in prepandemic community adolescents reported by Pace et al. [22] was 67% secure and 33% insecure, of which 23% were Ds, 7% were E, and 3% were D. Comparing pre-pandemic data with the current one, no significant differences in insecure–insecure classifications emerged in T1 or T2 (*X*^2^(1)_T1_ = 1.55 and *X*^2^(1)_T2_ = 0.21, both *p* > 0.05). 

### 3.2. Attachment Stability: FFI Test–Retest Reliability over Four Months

Respecting the assumptions (see Section 2.2.1), the analyses for test–retest reliability were performed. There was high concordance on two-way attachment categories (secure vs. all insecure ones, mostly dismissing) assigned in T1 and T2 (93.5% concordance, *k* = 0.84), suggesting attachment stability.

Regarding scorings on FFI dimensional scales, Kendall’s Tau-b bivariate correlations between FFI scores in T1 and T2 are reported in Table 1, revealing significant correlations between each pattern score in T1 and T2—i.e., in scores of S/F (*p* < 0.001), Ds (*p* < 0.001), P/E (*p* = 0.029), and D (*p* = 0.008)—and even between scores on the most relevant scale of overall narrative coherence (*p* < 0.001).

The analyses of discrepancies revealed that six participants secure in T1 were categorized as insecure in T2, four Ds, and two P/E, and one participant D in T1 obtained an organized Ds classification in T2.

Table 1 also reveals meaningful positive and negative correlations among the FFI scores for the four FFI attachment patterns with FFI overall coherence. For example, Table 1 shows that FFI coherence at T1 and T2 is significantly and positively linked up with the secure FFI score at T1 and T2, as well as significantly and negatively correlated with the FFI dismissal score at T1 and T2.

### 3.3. Relationships between Attachment in T1 and Psychotic-like Symptoms in T2

First, T2 thought problems’ scores, possibly linked to FFI category scores at T1, were checked. Table 1 shows that thought problems at T2 were linked up with attachment disorganization scores at T1 (0.24, *p* = 0.010). Also, Table 1 shows that Preoccupied attachment at T2 linked up with thought problems at T2 (0.22, *p* = 0.015).

## 4. Discussion

The main goals of this study were to evaluate the stability of adolescents’ attachment patterns over four months and examine longitudinal relationships between attachment patterns and psychotic-like symptoms in non-clinical adolescents through an interview.

Results for the first goal revealed a highly significant four-month stability of secure–insecure classifications among the adolescents in the Friends and Family Interview, suggesting that attachment patterns remained stable over four months, supporting past findings in community samples from Germany [59] and Italy [28], assessed with other attachment methods. In this study, attachment categories remained strongly stable despite the involvement of numerous interviewers, two coders, and the administration of interviews via video calls. As well, scores assigned to the same pattern in T1 And T2 were significantly related to each other. Overall, results support the test–retest reliability of the FFI over a brief period of four months. Thus, the FFI, like the other interviews like the AAI, CAI, and AICA, seems to be adequate in this psychometric property [27,28,29].

Moreover, in this group of community adolescents, attachment patterns remained steady throughout the potentially distressing COVID-19 pandemic phase, suggesting that these adolescents’ attachment representations might already be sufficiently formed to be unaffected by stressful circumstances, including being forced into self-isolation and social distancing due to pandemic-related limitations imposed by law [60,61]. The distribution of attachment categories among participants showed no difference from the distribution of the general adolescent population assessed with the same interview before [21].

Concerning the second goal, longitudinal relationships between adolescents’ attachment and PLEs were investigated. Given the stability of patterns, only the first measurement of attachment was considered concerning scores of PLEs four months later. Contrary to expectations based on previous findings with adolescents [30,48,49], adolescents categorized as secure or insecure did not differ in PLE levels. Concerning disorganization, there were too few participants categorized as disorganized to perform a comparison based on the organized–disorganized classification. However, the FFI system allows capturing levels of different types of insecurity within each participant, capturing sub-threshold levels of each pattern of insecurity even when they do not reach sufficient levels to assign a certain classification as primary, e.g., dimensional levels of preoccupation within an adolescent classified as secure. Thanks to these scores, it was possible to investigate the longitudinal relationships of all four attachment patterns based on four-point dimensional scores with PLEs. In this study, there were no relationships between scores for dismissing or preoccupied patterns and PLEs, as has been demonstrated in other studies relying on questionnaire-based studies with adolescents [32,48,49]. Yet there was a significant link between higher scores for attachment disorganization and more psychotic-like symptoms four months later. From an attachment perspective, this result adds some new information to the scarce literature connecting adolescents’ PLEs with their attachment patterns, highlighting a possible role of attachment disorganization reported here for the first time. In these non-clinical community adolescent participants, higher attachment disorganization was related to higher levels of dissociative experiences [47], extending to this age previous findings of the attachment literature on adults [45,46] and clinical insights coming from the research with the AAI in psychiatric adolescent inpatients [16]. Specifically, adolescents who lacked an organized attachment strategy guiding their relational behavior and expectations within significant relationships reported higher levels of symptoms captured by the thought problems scale of the YSR four months later, namely psychotic-like symptoms both of positive (i.e., hallucinations and bizarre behaviors) and negative nature (i.e., losing interest and motivation in life and lack of concentration), and dissociation-related experiences in forms typical in adolescence such as obsessive–compulsive symptoms, suicidality, and self-harming behaviors. From a clinical perspective, this result extends previous findings of attachment cross-sectional studies involving PLEs employing attachment self-report questionnaires [32,48,49], and calls for further investigation on the nature of the relationship found with attachment disorganization to understand if this form of insecurity would increase the adolescents’ vulnerability to PLE as it does with internalizing and externalizing problems [15].

### Limitations

This study has some strengths, namely the longitudinal design. It is the first study evaluating the test–retest reliability of the Friends and Family Interview, the first study investigating the longitudinal relationships between attachment and adolescents’ PLEs, and for the first time employs an attachment interview and not a questionnaire in doing it, which has allowed to detect the role of disorganization. Further, it is also the first study to administer attachment interviews during the lockdown due to the COVID-19 pandemic.

However, this study also has several limitations that preclude the generalizability of the results: First, the small number of individuals categorized as Preoccupied or Disorganized, precluded the possibility of testing the stability of the three-way and organized–disorganized categories and investigating the variation in PLEs according to the attachment category. Given that the FFI distributions in T1 and T2 reflected the distributions of the general pre-pandemic population of adolescents [22], this impossibility is probably due to the involvement of a community sample. This limit could be overcome by future studies with larger community samples or involving at-risk and clinical samples where the prevalence of preoccupied and disorganized categories is higher [16,62]. Moreover, the not-normal distribution of the data precluded the use of parametric analyses, calling for more research in larger samples where the data is more likely to be normally distributed. Second, the FFIs were conducted via video call due to pandemic-related social restrictions, and a replication of the study administering FFIs in person would extend the results’ generalizability to different modalities of assessment, including the traditional in-person administration forecast in the manual. Third, the disorganization was rated through the narrative scales applied to the FFI’s transcripts, not relying on the complete rating system of the FFI that includes rates of video signs of distress. Fourth, despite being suggested as reliable in assessing PLEs [51], the thought problems scale of the YSR assesses a wide range of symptoms, so future replication of the study employing multiple or specific measures of dissociation is recommended, eventually investigating the mediation and moderation of comorbid symptoms, e.g., the role of PLEs in the relationship between attachment and internalizing and externalizing symptoms and vice versa.

## 5. Conclusions

The present study strengthens the theoretical framework that postulates the stability of attachment patterns in adolescence, showing their continuity even in distressing periods such as the COVID-19 pandemic and its related lockdown. Additionally, this paper has highlighted that adolescents showing attachment disorganization also show more psychotic-like experiences, calling for future studies investigating the nature of this link, i.e., a long-term consequence of infant disorganization [47], normative adolescent attachment restructuration [7], and/or a reaction to the relational stress elicited by the COVID-19 pandemic [60,61]. Altogether, these results may reinforce the importance of attachment-oriented screening at the community level, something that could take place at high schools in the nurse’s or counselor’s office. This requires two assumptions already highlighted in the text: First, continue to make efforts towards the development of adequate and reliable tools for the assessment of attachment in adolescence. Second, the need for a common attachment background among all adults who interface with adolescents at different levels, i.e., parents and other relatives but also teachers, educators, sports coaches, doctors, psychologists, etc. This implies designing community interventions that allow all these adults to acquire a basic but common skill with respect to attachment theory, which would allow them to share an attachment lens with which to engage in dialogue among all stakeholders concerning adolescent well-being, so that efforts may be maximized to prevent mental ill-health and promote the well-being of adolescents. In this regard, although attachment theorists have been highlighting the need to bridge the gap between research and attachment-oriented clinics [63] and develop promising interventions [64], there’s still much to do to develop community-based attachment-oriented prevention. For instance, it could be helpful to provide schoolteachers [65] with guidelines on how to recognize situations at risk for attachment disorganization or to ask a professional for an attachment-based screening in case of signs of PLEs (e.g., in the case of one schoolmate referring that another teenager cuts him/herself or hears voices). An attachment lens at school can also be useful in stepping back and trying to understand the attachment needs behind apparently problematic behaviors of adolescents [66], being able to respond to them in a more sensitive and thoughtful way, e.g., by sharing and building knowledge with parents and relatives [67], promoting positive peer relationships to favor alternative attachment bonds beyond the family [68], and/or being able to suggest trauma- and attachment-informed assessment [69] and intervention for adolescents showing clinically severe levels of psychotic-like symptoms or their parents.

## Figures and Tables

**Table 1 ijerph-20-06562-t001:** Descriptives and Kendall Tau-b correlations among adolescents’ attachment patterns ^a^ in T1 and psychotic-like symptoms ^b^ in T2.

Variable	*M*	*SD*	1.	2.	3.	4.	5.	6.	7.	8.	9.	10.	11.
1. T1 FFI-S/F ^a^	3.11	0.81	-										
2. T1 FFI-Ds ^a^	1.68	0.81	−0.73 **	-									
3. T1 FFI-P/E ^a^	1.17	0.36	−0.19 *	−0.15	-								
4. T1 FFI-D ^a^	1.05	0.24	−0.21 *	0.07	0.16	-							
5. T1 FFI-Coh ^a^	3.33	0.58	0.74 **	−0.62 **	−0.12	−0.21	-						
6. T2 FFI-S/F	2.55	0.89	**0.53 ****	−0.56 **	0.09	−0.15	0.47 **	-					
7. T2 FFI-Ds ^a^	1.87	0.83	−0.47 **	**0.58 ****	−0.20 *	0.06	−0.43 **	-.67 **	-				
8. T2 FFI-P/E ^a^	1.31	0.45	−0.06	−0.05	**0.21 ***	0.24 *	−0.01	−0.20 *	−0.14	-			
9. T2 FFI-D ^a^	1.01	0.11	−0.20 *	0.15	0.10	**0.27 ****	−0.16	−0.18	0.20 *	0.21 *	-		
10. T2 FFI-Coh ^a^	2.92	0.50	0.43 **	−0.50 **	0.05	−0.15	**0.39 ****	0.73 **	−0.59 **	−0.13	−0.18	-	
11. T2 YSR-thought	1.05	1.51	−0.11	0.15	0.02	**0.24 ***	−0.08	−0.13	0.05	0.22 *	0.18	−0.06	-

Note. Statistical significance with * *p* < 0.05 * and ** *p* < 0.01. Correlations can be considered as weak (≥0.06), moderate (≥0.26), strong (≥0.49), very strong (≥0.71) [58]. Bold correlations are those pertinent to the aims of the research. The longitudinal study phases occurred during the COVID-19 pandemic, T1 = April–June 2020 (N = 102); T2 = August–October 2020 (N = 93). ^a^ Friends and Family Interview—S/F = secure autonomous; DS = insecure-dismissing; P/E = insecure-preoccupied; D = insecure-disorganized. Coh = overall narrative coherence. ^b^ Youth Self-Report 11–18 years-thought problems scale.

## Data Availability

The data presented in this study are available on request from the corresponding author. The data are not publicly available due to institutional policies.

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
