# Peer review of "Attachment Stability and Longitudinal Prediction of Psychotic-like Symptoms in Community Adolescents over Four Months of COVID-19 Pandemic"

_ijerph, 2023, doi:10.3390/ijerph20166562_

Round 1

Reviewer 1 Report

This research used the Friends and Family Interview (FFI) to study attachment stability and attachment-related psychopathological processes in adolescence. Test-retest reliability of this interview was followed, applied on a sample of 102 community adolescents assessed twice, during a severe COVID-19-related lockdown, at two points in time. Measures were attachment patterns, such as secure-autonomous, insecure-dismissing, insecure-preoccupied and insecure-disorganized. Moreover, the Thought Problems scale of the Youth Self Report to assess psychotic-like symptoms was implemented. The results revealed high stability of attachment classifications over four months (93.5%) and the predicted link between attachment disorganization (20%) at T1 with higher levels of psychotic–like symptoms of community adolescents at T2. Comments on FFI scale are provided along the conclusion about the relevant implications.

The paper presents an interesting work via quantitative research on the implementation of FFT and, in my opinion, succeeds to answer adequately to the research questions. The theoretical part is well-presented, introducing the reader to the theme.

The empirical part is well performed and provides all information needed to explicate the findings, while conclusions are clear.

  However, while reference to the reliability measures is provided, it lacks a corresponding account for validity. Thus, I recommend addressing this point and present some elements related to validity of the instruments implemented, which will enhance the methodological rigor of this work.

The manuscript needs minor language editing.

Author Response

Comments and Suggestions for Authors

This research used the Friends and Family Interview (FFI) to study attachment stability and attachment-related psychopathological processes in adolescence. Test-retest reliability of this interview was followed, applied on a sample of 102 community adolescents assessed twice, during a severe COVID-19-related lockdown, at two points in time. Measures were attachment patterns, such as secure-autonomous, insecure-dismissing, insecure-preoccupied and insecure-disorganized. Moreover, the Thought Problems scale of the Youth Self Report to assess psychotic-like symptoms was implemented. The results revealed high stability of attachment classifications over four months (93.5%) and the predicted link between attachment disorganization (20%) at T1 with higher levels of psychotic–like symptoms of community adolescents at T2. Comments on FFI scale are provided along the conclusion about the relevant implications.

The paper presents an interesting work via quantitative research on the implementation of FFT and, in my opinion, succeeds to answer adequately to the research questions. The theoretical part is well-presented, introducing the reader to the theme. The empirical part is well performed and provides all information needed to explicate the findings, while conclusions are clear.

R.: We are grateful to Reviewer1 for his/her appreciation and comments.

However, while reference to the reliability measures is provided, it lacks a corresponding account for validity. Thus, I recommend addressing this point and present some elements related to validity of the instruments implemented, which will enhance the methodological rigor of this work.

R.: Sorry for this missing information. We have expanded prior information as follows: “Specifically, the validity of the FFI has been demonstrated in different studies [17, 20-26]. Convergent validity has been supported by studies that found that FFI scores/classifications responses in 11-year-old and over children were predicted by pre-birth [17] and contemporary [20] parents’ Adult Attachment Interview (AAI) responses. As well, FFI outcomes were predicted by SSP data of the same adolescent participants when infants [17], and by a study that showed that the FFI distribution overlapped with the AAI meta-analytical distributions in community adolescents’ populations [21]. Further, the coherence in the FFI showed discriminant validity from middle-aged children's verbal intelligence [17], and patterns and coherence resulted independent of emotional-behavioral problems and cognitive-verbal abilities of adolescents [21], as well as EEG responses [22]. Moreover, the FFI shows concurrent validity with attachment questionnaires [23, 24], content validity with a factorial structure [25], and adequate inter-country consistency in Belgium and Romania [26].”

For this reason, the following references were added:

  • Escobar MJ, Santelices MP (2013). Attachment in adopted adolescents. National adoption in Chile. Child Youth Serv Rev, 2013, 35(3), 488-492. https://doi.org/10.1016/j.childyouth.2012.12.011
  • Muzi S, Pace CS, Steele H. The friends and family interview converges with the inventory of parent and peer attachment in community but not institutionalized adolescents. J Child Family Studies 2022.https://doi.org/10.1007/s10826-021-02181-1
  • Psouni E, Apetroaia A. (2014). Measuring scripted attachment-related knowledge in middle childhood: The Secure Base Script Test. Attach hum develop, 16(1): 22-41. https://doi.org/10.1080/14616734.2013.804329

Comments on the Quality of English Language                      

The manuscript needs minor language editing.

R.: Thank you, the manuscript has been carefully and entirely proofread and checked by a native English speaker (last co-author).

Reviewer 2 Report

Unclear used tests - can not be performed the parametric test (t-test, ANOVA, Reggression, Pearson) coef..  if assumptions are not tested.

Moreover, there is a huge disproportion in the number of subsamples that are compared (ANOVA). Homogeneity of variances is not tested; in Linear regression; linearity, multicollinearity, autocorrelation, and presence of outliers (Mahalanobis distance) are not tested, too.

Unclear used tests - can not be performed the parametric test (t-test, ANOVA, Reggression, Pearson) coef..  if assumptions are not tested.

Moreover, there is a huge disproportion in the number of subsamples that are compared (ANOVA). Homogeneity of variances is not tested; in Linear regression; linearity, multicollinearity, autocorrelation, and presence of outliers (Mahalanobis distance) are not tested, too.

Author Response

Comments and Suggestions for Authors

Unclear used tests - can not be performed the parametric test (t-test, ANOVA, Regression, Pearson) coef..  if assumptions are not tested.

Moreover, there is a huge disproportion in the number of subsamples that are compared (ANOVA). Homogeneity of variances is not tested; in Linear regression; linearity, multicollinearity, autocorrelation, and presence of outliers (Mahalanobis distance) are not tested, too.

R.: We thank Reviewer 2 for these observations that have allowed has to improve the quality of our work. Following the Reviewer’s 2 suggestions, we have checked and managed the presence of outliers, the homogeneity of variances, and the normality of distribution. Given we found that data were not normally distributed, we did not test the linearity, multicollinearity, and autocorrelation as we have decided to use non-parametric tests. The results for the first aim did not change, while for the second aim, we have removed the regression relying on the Kruskal-Wallis test (not significant) and Kendall’s tau-b correlations, so the discussion has been modified consequently.

Specifically, we have made the following revisions:

In the method section (2.3)
“The FFI categories distribution and descriptives (Mean [M] and Standard Deviation [SD]) have been reported for all measures, with statistically significance for all hypotheses was set with the tolerable limit of type-one errors being set at the conventional p <.05.

After checking and managing outliers according to the criteria suggested by Hoaglin et al. [56], the two main assumptions for the use of parametric tests were test-ed, i.e., homogeneity of variances (by Levene’s tests) and normality of the distribution (by Shapiro-Wilk tests). Given data were not normally distributed, non-parametric tests were employed [57] as they do not require the assumption of normality of the population from which the sample is drawn, and they are appropriate for the analysis of dependent variables on dichotomous, categorical, and ordinal scales, as in this pilot study.

As preliminary analyses, inter-rater agreement on the FFI classification was checked through Cohen’s k, and chi-square tests were used to check the effect of the interviewer (see above, 2.2.1) and of participants’ gender on FFI secure-insecure categories distributions in T1 and T2, as well as to compare such distributions to the pre-pandemic one [22] as indicative of the general population distribution. The U-Mann Whitney test was used to compare the age of participants classified as secure or insecure, and the Kruskal-Wallis test to check differences among four categories, employing Kendall’s tau-b correlation coefficient (τb) to check age differences in T2 YSR/though problems score.

For the first aim, the test-retest reliability of the FFI categories was tested using k di Cohen to verify the association between T1 and T2 FFI secure-insecure distribution, and τb was used to inspect relationships between patterns and coherences scores in the two data collections, with the effect considered as: ≥ 0.06 weak, ≥ 0.26 moderate, ≥ 0.49 strong, ≥0.71 very strong [58].

For the second aim, the effect of the FFI category on YSR/thought problems’ scores in T2 was checked through the U-Mann-Whitney test (two-way and organized-disorganized systems) and the Kruskal-Wallis H test for the four-way system, also checking Kendall’s tau-b correlations between FFI patterns and coherence scores in T1 and YSR/thought problems in T2.”

- In the results section and Table 1 of correlations

In 3.1Means and Standard Deviations for all measures are reported in Table 1 (below).

Given the paucity/absence of FFI transcripts assigned to P and/or D categories in both T1 and T2, analyses on three and four-way distribution could not be performed (i.e., X2 comparisons and Kruskal-Wallis test), and following analyses on categorical data were performed considering the FFI secure-insecure (including Ds, P, and D) distribution.

Gender and age differences. No gender differences were revealed in FFI secure-insecure categories distribution in T1 or T2 through the chi-square test, p >.071, but males showed lower scores for security than girls both in T1, U = 1643.5, p = .014, and in T2, U = 1296, p = .040, also showing higher scores in the dismissing pattern at T1, U = 965.5, p = .022.

Participants classified as secure were older than insecure ones in T1, U = 634.5, p = .033, and T2, U = 587.5, p = .013.

The T2 YSR/Thought problems score was not correlated to gender (p = .38) or age (p = .31). 

Correspondence with pre-pandemic FFI data. The FFI categories distribution in pre-pandemic community adolescents reported by Pace et al. [22], was 67% secure and 33% insecure, of which 23% Ds, 7% E, and 3% D. Comparing pre-pandemic data with the current one, no significant differences in insecure-insecure classifications emerged in T1 or T2 (X2(1)T1 = 1.55 and X2(1)T2 = 0.21, both p > .05).”

In 3.2 “Regarding scorings on FFI dimensional scales, Kendall’s Tau-b bivariate correlations between FFI scores in T1 and T2 are reported in Table 1, revealing significant correlations between each pattern score in T1 and T2 -i.e., in scores of S/F (p < .001), Ds (p <.001), P/E (p =.029) and D (p = .008)- and even between scores on the most relevant scale of overall narrative coherence (p <.001).” […]

Table 1 also reveals meaningful positive and negative correlations among the FFI scores for the four FFI attachment patterns with FFI overall coherence. For example, Table 1 shows that FFI coherence at T1 and T2 is significantly and positively linked up with the secure FFI score at T1 and T2, as well as significantly and negatively correlated with FFI dismissal score at T1 and T2.”

In 3.3. “First, T2 Thought Problems’ scores possibly linked to FFI category scores at T1 were checked.  Table 1 shows that Thought Problems at T2 were linked up with attachment Disorganization scores at T1 (.24, p = .010). Also, Table 1 shows that Preoccupied attachment at T2 linked up with Thought Problems at T2 (.22, p =.015).”

The abstract, discussion and limitations were revised accordingly, e.g.Moreover, the not-normal distribution of the data precluded the use of parametric analyses, calling for more research in larger samples where data could be more likely normally distributed.”

For this reason, we have added the following references

  1. Hoaglin D C., Mosteller, F., and Tukey, J. W. (1983). Understanding Robust Exploratory Data Analysis. Wiley, New York. (Good introduction to robust statistics.)
  2. Kendall M, Gibbons JD. Rank Correlation Methods. 5th Edition, 1990, Edward Arnold, London.
  3. Schober P, Boer C., Schwarte LA. Correlation Coefficients: Appropriate Use and Interpretation. Anesthesia & Analgesia, 2018, 126(5): 1763-1768. https://doi.org./10.1213/ANE.0000000000002864

Moderate editing of English language required.

R.: Thank you, the manuscript has been carefully and entirely proofread and checked by a native English speaker (last co-author).

Reviewer 3 Report

There are some aspects that I feel require some addressing.

1) The literature review is generally clear. Some careful proofreading will be required for minor errors.

2) I feel that a section 1.3 could be added to the review, summarising both goals as well as stating any hypotheses that were evaluated.

3) The methodology section is clear in content but will require more careful proofreading (English language) than the literature review.

4) P. 5 refers to a "total sample" in contrast to T1 (line 209). It is unclear what this "total sample" refers to. The original 125? Why would there be demographic data for them?

5) Table 1 is quite space consuming and I wonder if it actually contributes to understanding the results (other than age and gender)?

6) As it is later pointed out as a limitation, is the distribution across the four attachment patterns reflective of the population in general?

7) The conclusion is very short. I would like to see a little more in relation to the potential community based implications of the analysis that is hinted at.

8) It will make sense to have one person proofread the entire text as different sections were clearly written by different authors, and so there is sometimes inconsistency in use of abbreviations etc.

Please see my comments and suggestions where I make reference to language.

Author Response

Comments and Suggestions for Authors

There are some aspects that I feel require some addressing.

1) The literature review is generally clear. Some careful proofreading will be required for minor errors.

R.: Thank you, the manuscript has been carefully and entirely proofread and checked by a native English speaker (last co-author).

2) I feel that a section 1.3 could be added to the review, summarising both goals as well as stating any hypotheses that were evaluated.

R.: Thank you for this useful suggestion. We have added the following section:

“1.3 The current study.

From the above summary of published work including the Friends and Family Interview or FFI, some notable gaps are evident in the FFI literature.  First, the recent effort of researchers in contributing to the knowledge about the psychometric properties of the FFI [17-26] has not yet addressed the topic of the test-retest validity of this interview, of which investigation would also increase the knowledge about processes of continuity or discontinuity in attachment during the critical adolescent period of development. Therefore, the first goal of this study is to investigate the stability of adolescents’ attachment representations assessed by measuring the test-retest reliability of the FFI categories over four months, hypothesizing stability of attachment classification in this brief period as in other studies employing other interviews with adolescents [27-29].

Second, the dramatic increase in PLEs of adolescents after the COVID pandemic calls for further investigation efforts on possible correlates of these potentially psychopathological symptoms, and so far the attachment literature has suggested that attachment insecurity may be related to dissociative experiences in adults [42-45] and PLEs in adolescents [30, 45, 46], but longitudinal and interview-based studies are still few in number, and no studies investigated associates of disorganization in response to the FFI. Therefore, a second goal of this study is to examine the longitudinal associations between attachment patterns secure, insecure dismissing, insecure-preoccupied, and insecure-disorganized when compared and PLEs reported on by non-clinical community adolescents, and the FFI will permit consideration of different types of insecure patterns providing a more comprehensive picture of adolescent adaptation than is available from questionnaires [15]. Based on existing findings on adolescents, higher levels of PLEs were expected to be reported from those adolescents with higher scores for insecure (preoccupied or dismissing) patterns, and especially for those adolesacents with higher disorganization scores in response to the FFI. Such expectations are consisted with past findings re PLEs concerning adults [32, 41-49].”

In this regard, we have added the following references:

  • Escobar MJ, Santelices MP (2013). Attachment in adopted adolescents. National adoption in Chile. Child Youth Serv Rev, 2013, 35(3), 488-492. https://doi.org/10.1016/j.childyouth.2012.12.011
  • Muzi S, Pace CS, Steele H. The friends and family interview converges with the inventory of parent and peer attachment in community but not institutionalized adolescents. J Child Family Studies 2022. https://doi.org/10.1007/s10826-021-02181-1
  • Carlson EA. A prospective longitudinal study of attachment disorganization/disorientation. Child Dev. 1998 Aug;69(4):1107-28.

to the reference list.

3) The methodology section is clear in content but will require more careful proofreading (English language) than the literature review.

R.: Thank you, the manuscript has been proofread by a native English speaker.

4) P. 5 refers to a "total sample" in contrast to T1 (line 209). It is unclear what this "total sample" refers to. The original 125? Why would there be demographic data for them?

R.: We thank Reviewer 3 to allow us to notice this error. That referred to differences between the sample completing T1 and the subsample completing T2. We have modified it as follows “Inspection of demographic data was undertaken to see if there were differences between the sample in T1 and the subsample who completed T2, and there were no significant differences in gender or family SES distribution, or age (all p > .152).

5) Table 1 is quite space consuming and I wonder if it actually contributes to understanding the results (other than age and gender)?

R.: We agree and We have removed Table 1 by inserting relevant information about participants in the text, as follows:

“this study T1 includes 102 community adolescents aged 12-17 years old (45% males; Mean age [Mage] = 14.64, Standard Deviation [SDage] = 1.63), coming from low-risk fami-lies of medium (77%) or high (12%) social educational level.

6) As it is later pointed out as a limitation, is the distribution across the four attachment patterns reflective of the population in general?

R.: Thank you for this question that has allowed us to be more accurate in presenting our results. We have added a comparison with the distribution of general population, reported in the result section as follows “Correspondence with pre-pandemic FFI data. The FFI categories distribution in pre-pandemic community adolescents reported by Pace et al. [22], was 67% secure and 33% insecure, of which 23% Ds, 7% E, and 3% D. Comparing pre-pandemic data with the current one, no significant differences in insecure-insecure classifications emerged in T1 or T2 (X2(1)T1 = 1.55 and X2(1)T2 = 0.21, both p > .05).

And commented in the discussion as follows “Moreover, in this group of community adolescents, attachment patterns remained steady throughout the potentially distressing COVID-19 pandemic phase, suggesting that these adolescents' attachment representations might be already sufficiently formed to be unaffected by stressful circumstances, including being forced into self-isolation and social distancing due to pandemic-related limitations imposed by law [60, 61]. The distribution of attachment categories of participants showed no difference with the distribution of the general adolescent population assessed with the same in-terview before the, indeed [21].

7) The conclusion is very short. I would like to see a little more in relation to the potential community-based implications of the analysis that is hinted at.

R.: Thank you to allow us to expand the discussion about potential community-based implications of the analyses conducted. First, we must clarify that, according to the revisions suggested by Reviewer 2, the results for the second goal slightly changed, and the discussion accordingly. We have expanded the conclusions as follows:

Additionally, this paper has highlighted that adolescents showing attachment disorganization also show more psychotic-like experiences, calling for future studies investigating the nature of this link, i.e., a long-term consequence of infant disorganization [47], and/or normative adolescent attachment restructuration [7] and/or reaction to the relational stress elicited by the COVID-19 pandemic [60,61]. Altogether, these results may reinforce the importance of attachment-oriented screening at a community level, something that could take place at high schools in the nurse’s or counselor’s office. This requires two assumptions already highlighted in the text: First, continue to make efforts in the development of adequate and reliable tools for the assessment of attachment in adolescence. Second, the need of a common attachment background among all adults who interface with adolescents at different levels, i.e., parents and other relatives but also teachers, educators, sports coaches, doctors, psychologists, etc. This im-plies designing community interventions that allow all these adults to acquire a basic but common skill with respect to attachment theory, which would allow them to share an attachment lens with which to engage in dialogue among all stakeholders’ concerning adolescent wellbeing, so that efforts may be maximized to prevent mental ill-health and promote the well-being in adolescents. In this regard, although for some time now attachment theorists have been highlighting the need to bridge the gap be-tween research and attachment- oriented clinics [63], developing promising interventions [64], there’s still much to do to develop community-based attachment-oriented prevention. For instance, it could be helpful providing schoolteachers [65] with guidelines on how to recognize situations at risk for attachment disorganization, or to ask a professional with an attachment-based screening in case of signs of PLEs (e.g., in the case of one schoolmate refer that other teenagers cut him/herself or listen voices). An attachment lens at school can be also useful in stepping-back and trying to under-stand the attachment needs behind apparently problematic behaviors of adolescents [66], being able to respond them in a more sensitive and thoughtful way, e.g., sharing and building knowledge with parents and relatives [67], promoting positive peer relationships to favor alternative attachment bonds beyond the family [68] and/or being able to suggest trauma- and attachment-informed assessment and intervention for adolescents showing clinically-severe levels of psychotic-like symptoms or their parents

Adding these references:

  • Oppenheim D, Goldsmith DF (Eds.). Attachment theory in clinical work with children: Bridging the gap between research and practice. 2011. Guilford press.
  • Steele H, Steele M (Eds.). Handbook of attachment-based interventions. 2017. Guilford Publications.
  • Riley, P. (2010). Attachment theory and the teacher-student relationship: A practical guide for teachers, teacher educators and school leaders. Routledge. https://eclass.edc.uoc.gr/modules/document/file.php/DEA104/Attachment%20Theory%20Teacher-Student%20Relationship_%20A%20Practical%20Guide%20for%20Teachers%2C%20Teacher%20Educators.pdf (last access 03/07/2023).
  • Barone L, Carone N, Genschow J, Merelli S, Costantino A. (2020). Training parents to adolescents' challenges: the CONNECT parent program. Studi e ricerche: 31-46. https://doi.org/10.3280/qpc46-2020oa10160
  • Magaldi-Dopman D, Conway T. Allied Forces: The Working Alliance for Meaningful Parent-Educator Partnerships in Special Education. Journal of Special Education Apprenticeship 2012, 1(2): n2. https://files.eric.ed.gov/fulltext/EJ1127909.pdf
  • Muzi S, Rogier G, Pace CS. Peer power! Secure peer attachment mediates the effect of parental attachment on depressive withdrawal of teenagers. Int j env res pub health, 2022, 19(7): 4068. https://doi.org/10.3390/ijerph19074068

8) It will make sense to have one person proofread the entire text as different sections were clearly written by different authors, and so there is sometimes inconsistency in use of abbreviations etc.

R.: Thank you, the manuscript has been proofread by a native English speaker that checked for the consistency of the writing style as well.

Comments on the Quality of English Language

Please see my comments and suggestions where I make reference to language.

R.: Thank you, see previous responses. The manuscript has been carefully and entirely proofread and checked by a native English speaker (last co-author).
